# PipeOffload: Improving Scalability of Pipeline Parallelism with Memory Optimization

**Xinyi Wan** [* 1 2]  **Penghui Qi** [* 1 2]  **Guangxing Huang** [1]  **Min Lin** [1]  **Jialin Li** [2]

## Abstract

Pipeline parallelism (PP) is widely used for training large language models (LLMs), yet its scalability is often constrained by high activation memory consumption as the number of in-flight microbatches grows with the degree of PP. In this paper, we focus on addressing this challenge by leveraging the under-explored memory offload strategy in PP. With empirical study, we discover that in the majority of standard configurations, at least half, and potentially all, of the activations can be offloaded with negligible overhead. In the cases where full overload is not possible, we introduce a novel selective offload strategy that decreases peak activation memory in a better-than-linear manner. Furthermore, we integrate memory offload with other techniques to jointly consider overall throughput and memory limitation. Our experiments proves that the per-device activation memory effectively reduces with the total number of stages, making PP a stronger alternative than TP, offering up to a 19% acceleration with even lower memory consumption. The implementation is open-sourced at this url.

## 1. Introduction

As modern large transformer models (Vaswani et al., 2017) scale towards trillions of parameters, model parallelism becomes essential for distributing model parameters across multiple devices. Compared to ZeRO (Rajbhandari et al., 2020) and tensor parallelism (Shoeybi et al., 2019), pipeline parallel (PP) (Huang et al., 2019; Harlap et al., 2018) has a lower communication volume and a higher arithmetic intensity. However, while PP shards layers across devices to reduce parameter memory requirements, its scalability remains constrained by the activation memory. Increasing the number of PP stages reduces layers per device but necessitates more in-flight microbatches to minimize pipeline bubbles. This trade-off leaves overall activation memory demands unchanged.

In this work, we address this memory limitation of PP by offloading memory to the host. While memory offload is widely adopted in data parallelism (DP) (Goyal et al., 2017; Ren et al., 2021), its potential in PP remains largely unexplored. PP is particularly suited for memory offload because the gap between the forward pass and the backward pass creates a natural window for offloading and reloading activation memory without interfering with other computations. This contrasts sharply with activation rematerialization (Chen et al., 2016) which introduces significant recomputation overhead. Memory offload, when properly scheduled and overlapped with other computation, can be a free lunch.

Formally, for a single transformer layer, if we define $T_o$ as the round-trip time for its activation memory to be moved from device to host (D2H) and then host to device (H2D), $T_c$ as the time for its total forward and backward compute, then the ratio between them, denoted as $k = T_o/T_c$, is an important indicator on the proportion of the activation memory that can be offloaded without introducing significant efficiency drawbacks. Full activation memory offload is possible if $k \leq 1$, as the offload operations can be fully overlapped by compute. Without considering the trivial layers which can be recomputed with negligible overhead (e.g. Dropout, GeLU, LayerNorm), we can estimate the offload cost for the rest of the layers and subsequently estimate $k$. With sequence length ($s$), hidden size ($h$), PCI-E duplex bandwidth ($B_o$) and GPU compute bandwidth ($B_c$), $k$ is computed as: (Narayanan et al., 2021; Korthikanti et al., 2023)

$$k = \frac{T_o}{T_c} = \frac{10}{3(6h + s)} * \frac{B_c}{B_o} \quad (1)$$

Note that the value of $k$ decreases as model size or sequence length increases. Figure 1 (left) demonstrates that the value of $k$ is surprisingly small under typical hidden dimension and sequence size settings, indicating that a significant portion of activation memory can be offloaded. We observe that $k \leq 1$ when the hidden dimension exceeds 8k or the

---

[*]Equal contribution  [1]Sea AI Lab  [2]National University of Singapore. Correspondence to: Min Lin <linmin@sea.com>, Jialin Li <lijl@comp.nus.edu.sg>.

*Proceedings of the 42nd International Conference on Machine Learning*, Vancouver, Canada. PMLR 267, 2025. Copyright 2025 by the author(s).

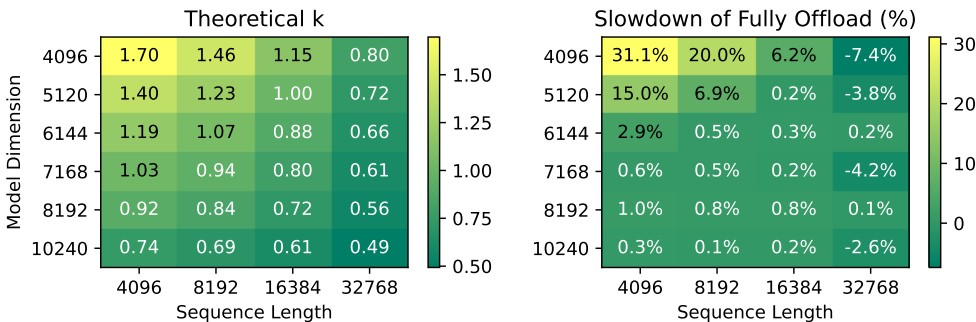

*Figure 1.* Ablation studies on offload overhead on NVIDIA A100 GPUs. On the left, $k$ values were estimated using Formula 1 with $B_c = 220$ TFLOPS/s and $B_o = 15$ GB/s. On the right, the reduction in throughput due to offload was measured through experiments. The experiments are performed utilizing the fully offloaded 1F1B schedule outlined in Figure A, involving 8 PP devices and 32 microbatches. The number of transformer layers was chosen to ensure that the baseline, without offload, does not OOM. It is important to note that some values in the second graph are negative, as the baseline experiences frequent CUDA malloc/dealloc operations due to high memory usage.

sequence length is greater than 16k; under these conditions, **all activation memory can be offloaded with negligible overhead**, as shown in Figure 1 (right).

When $k > 1$, offloading all activation memory would lead to a decrease in throughput. In this scenario, we resort to a partial offload where a subset of the activations are offloaded. Similar to the rematerialization approach where trivial operations are preferred as they impose less computation burden, we perform **selective offload** that prioritize activations that yield the greatest reduction in peak memory usage. We introduce a general guideline for selective offload that always prefers activations with longer lifespan, i.e., longer gaps between forward and backward passes. Intuitively, the longer an activation remains in flight, the more it contributes to peak memory.

A widely adopted strategy to reduce pipeline bubbles involves placing multiple stages on the same device, as demonstrated in works such as (Narayanan et al., 2021), (Qi et al., 2024), and (Liu et al., 2023). Notably, different stages have varying lifespans, potentially resulting in a more efficient, better-than-linear reduction in peak memory usage. The efficiency of selective offload also depends on the memory usage pattern of the pipeline schedule. For instance, when visualizing the memory usage patterns of interleaved 1F1B (Narayanan et al., 2021) and our *PipeOffload* method in Figure 2, it is apparent that offloading stage 0 in our method yields a 3/4 reduction in peak memory, while in the interleaved 1F1B, at most a 1/2 reduction can be achieved.

When memory offload is applied in practice, we need to further consider its interplay with other factors, especially the trade-off between memory and throughput. We extend the interleaving strategy into a generalized form, offering smooth memory reduction with minimal throughput loss. This approach provides flexibility in optimizing performance based on specific system needs.

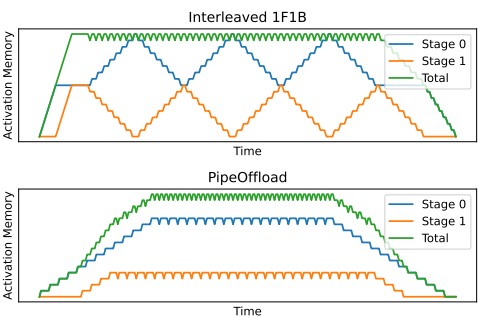

*Figure 2.* Memory pattern of different schedules. We plot the activation memory of each stage separately and show their contribution to the total activation memory. In Interleaved 1F1B, offloading stage 0 results in only a 50% reduction in peak activation memory, despite stage 0 having a longer lifespan. Contrastingly, *PipeOffload* at the bottom distributes activation memory uniformly across time, offering better-than-linear memory savings if stage 0 is offloaded.

In the rest of this paper, we detail the selective offload strategy in Section 2, introduce a family of pipeline schedules that trade-off between throughput and peak memory in Section 3, elaborate on implementation details in Section 4, and finally evaluate and compare the methods in Section 5.

## 2. Selective Offload

When full activation offload is not feasible ($k > 1$) without overhead on the throughput, an efficient selective offload strategy becomes crucial. In this section, we explore considerations for a selective offload strategy when multiple pipeline stages are placed into each device as in interleaved 1F1B (Narayanan et al., 2021).

Following (Qi et al., 2024), we decompose a pipeline schedule into repeating a building block, which describes how a

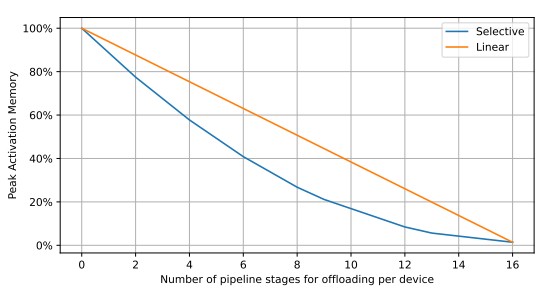

*Figure 3.* The memory reduction ratio of stage-level offload under 8 PP devices and 16 stages.

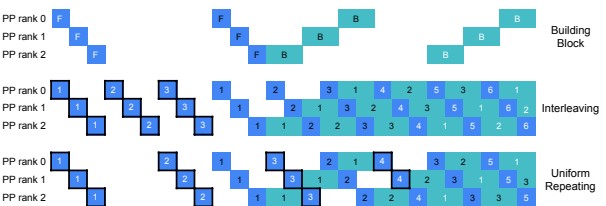

*Figure 4.* The building block (top) describes the pattern for each microbatch, where *F* represents forward and *B* represents backward. Both the interleaving (middle) and uniform repeating (bottom) strategies adhere to this building block. Although sharing the same peak memory, the contributions of pipeline stages differ in these two strategies. We emphasize the contributions of the first pipeline stages with bold borders.

single microbatch should be scheduled in the pipeline (as in Figure 4). By ensuring each microbatch adheres to this pattern, the peak memory usage is approximately proportional to the summed lifespan of each pipeline stage. Intuitively, the longer activations remain in flight, the more memory a PP schedule consumes. Based on this insight, preferring offloading pipeline stages with longer lifespan is a natural selective strategy for more memory reduction.

However, lifespan is not the sole factor influencing a stage's contribution to peak memory. The strategy for organizing microbatches within a pipeline schedule also significantly impacts how stages contribute to peak memory. We evaluate two strategies for building pipeline schedules: the interleaving strategy (as in (Narayanan et al., 2021)) and the uniform repeating strategy (as in (Qi et al., 2024)). The interleaving strategy employs a bi-level repeating pattern: the outer loop interleaves pipeline stages, while the inner loop uniformly repeats a set number of microbatches. In contrast, the uniform repeating strategy consistently schedules the next microbatch after a fixed offset. As shown in Figure 4, although both strategies share the same building block and peak memory, their contributions to peak memory differ. Specifically, offloading or recomputing the first pipeline stage (indicated by white numbers) in rank 0 reduces peak memory by 3 activations in the interleaving strategy, whereas the uniform repeating strategy achieves a reduction of 4 activations. This demonstrates a greater memory reduction with the uniform repeating strategy than with interleaving srategy.

Figure 2 illustrates the contribution of each pipeline stage to peak activation memory across 8 devices. It is evident that in both strategies, stage 0 (with longer lifespan) contributes more than or equal to stage 1 (with shorter lifespan) to peak memory. Notably, the uniform repeating strategy, where each stage's contribution to peak memory is roughly proportional to its lifespan (Qi et al., 2024), tends to offer greater memory reduction compared to the interleaving strategy under the same budget.

Based on these observations, we propose prioritizing the offloading of pipeline stages with longer lifespans. The uniform repeating strategy should be favored over the interleaving strategy for its superior memory reduction efficiency within the same offload budget. Figure 3 presents the theoretical peak memory curve under various offload budgets using the uniform repeating strategy. It demonstrates better-than-linear efficiency in memory reduction. Notably, by offloading only half of the pipeline stages, peak memory can be reduced to approximately one-quarter in scenarios with an 8 PP degree and 16 pipeline stages per device.

## 3. Trading off Memory and Throughput

In PP, there is a trade-off between activation memory and pipeline bubbles. While the uniform repeating strategy offers superior memory reduction efficiency through offload, it may also lead to an increased number of pipeline bubbles compared to the interleaving strategy, as illustrated in Figure 4. In scenarios where memory pressure is not a primary concern, the interleaving strategy is preferable due to its ability to maintain higher throughput. In this section, we extend the interleaving strategy into a generalized form. This approach aims to achieve smooth memory reduction while minimizing throughput loss, providing a flexible solution that can be tailored to specific system requirements.

### 3.1. Free Lunch for Interleaved 1F1B

**Zero-Bubble Strategy**    We use the zero-bubble strategy from (Qi et al., 2023) to reduce pipeline bubbles without any trade-offs. Specifically, the backward pass is split into activation gradient computation (*B*) and weight gradient computation (*W*). Unlike *ZB-H1* and *ZB-H2* schedules in (Qi et al., 2023), we don't delay *W* passes to further reduce bubbles, as this complicates our strategies. Since our focus is mainly on memory reduction, we simply keep their original schedules with split backward pass (see the middle of Figure 5).

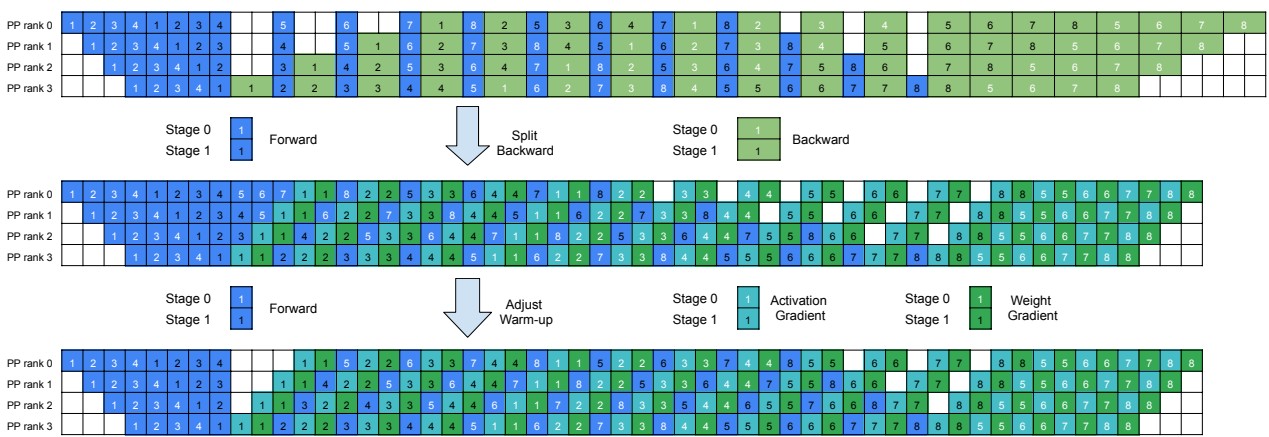

*Figure 5.* Top: vanilla interleaved 1F1B; Middle: with split backward; Bottom (*GIS*): after adjusting warmup.

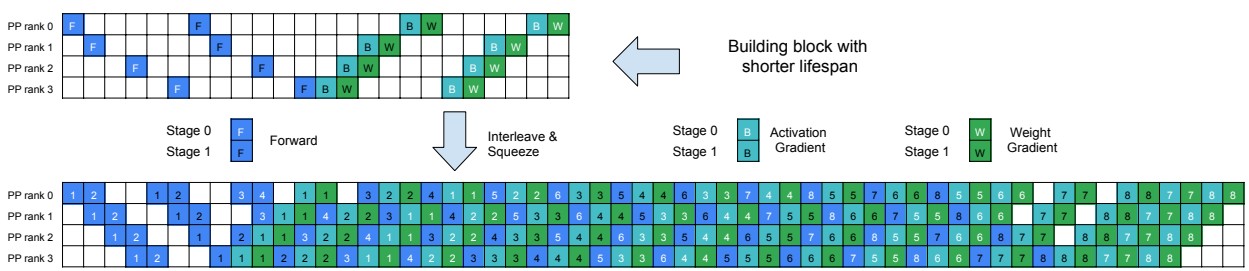

*Figure 6.* Top: the building block of *GIS-H*; Bottom: *GIS-H* schedule with less activation memory.

**Adjust Warmup** We modify the vanilla interleaved 1F1B schedule to make it more memory efficient, as shown in Figure 5. Specifically, we reduce the number of forward passes in the warmup phase from $d(v-1) + 2(d-i) - 1$ to $d(v-1) + d - i$ for PP rank $i$, where $d$ is the number of devices and $v$ is the number of pipeline stages per device. This change reduces the maximum peak memory (occurring in rank 0) from $dv + d - 1$ to $dv$, which is a prominent improvement especially for small $v$. Importantly, this modification achieves memory savings without introducing any additional pipeline bubbles.

For convenience, we refer to the schedule after splitting backward and adjusting warmup as *GIS*.

### 3.2. Memory Reduction by Shortening Lifespan

We generalize the interleaving strategy to accommodate smaller memory constraints with minimal efficiency loss, while maintaining a 1F1B pattern. Inspired by (Qi et al., 2024), we reduce peak memory usage by shortening the lifespan of the building block.

From Figure 5, we observe redundant lifespans for each microbatch. For instance, in *GIS*, although backward passes have tight dependencies within each stage, the waiting time between stages is excessively long. This presents an op-

portunity to further reduce activation memory. We design new building blocks with shortened lifespans (Figure 6) and organize microbatches into a pipeline schedule by repeating these building blocks. Similar to interleaved 1F1B, we use a bi-level repeating pattern: the outer loop interleaves pipeline stages, and the inner loop uniformly repeats a number of microbatches (denoted as $g$). Notably, *GIS* is a special case where $g$ equals $d$. We can adjust $g$ to control the lifespan, with a minimal value of $\lceil \frac{d}{2} \rceil$ to satisfy dependencies between stages. Note that reducing the lifespan incurs extra pipeline bubbles during the warmup phase. For any value of $g$ ($\lceil \frac{d}{2} \rceil \leq g \leq d$), the maximum peak activation memory is $g(v-1) + d$ (in rank 0), and the size of extra pipeline bubbles is $(d-g) * (v-1)$. By selecting the largest value of $g$ that fits within the memory limit, we can achieve an optimal schedule with minimal throughput loss. In the extreme case where $g = \lceil \frac{d}{2} \rceil$ (Figure 6), it results in peak memory usage that is about half of the vanilla interleaved 1F1B. We refer to this specific schedule as *GIS-H*.

By uniformly repeating the building block (as described in Section 2) of *GIS-H*, with minor modifications to prevent collisions (Qi et al., 2024), we present a schedule called *PipeOffload* with similar peak memory. In *PipeOffload*, offloading half of the pipeline stages is referred to as *PO-H*, while offloading all stages is termed *PO-F*. We compare the

bubble and activation memory in Table 1.

*Table 1.* Comparing activation memory and bubble rate of different schedules. Additional notations: Activation memory of the entire model ($M$), Time of *F*, *B*, *W* passes of a single stage on a device ($T_F, T_B, T_W$). Note that for *PO-F*, the activation memory proportionally decreases with $vd$.

| Schedule | Activation Memory | Bubble |
|----------|-------------------|--------|
| *1F1B* | $M$ | $v(d-1)(T_F + T_B + T_W)$ |
| *1F1B-I* | $\frac{(v+1)}{v}M$ | $(d-1)(T_F + T_B + T_W)$ |
| *GIS* | $M$ | $(d-1)(T_F + T_B)$ |
| *GIS-H* | $\frac{(v+1)}{2v}M$ | $(d-1)(T_F + T_B) + \frac{(v-1)d}{2}(T_F + T_B - T_W)$ |
| *PO-H* | $\approx \frac{(v+2)}{8v}M$ | $< v(d-1)(T_F + T_B + T_W)$ |
| *PO-F* | $O(\frac{M}{vd})$ | $< v(d-1)(T_F + T_B + T_W)$ |

## 4. Offload Implementation

In this section, we elaborate the implementation details of our offload strategy, particularly highlighting the differences with the approach described in (Yuan et al., 2024). As we aim to reduce memory usage with minimal overhead, we primarily focus on leveraging the "free lunch" opportunity of offload, which necessitates that the offloading and reloading processes can be fully overlapped with computation, thereby avoiding any additional overhead for the original pipeline.

### 4.1. Improve Offload Efficiency

To reduce the offload constraint and enable greater offload capabilities, we adopted the subsequent approaches: a) Employing direct recomputation on activation-heavy but computationally lightweight layers like GeLU to diminish activation memory per layer, leading to a notable 40% decrease in activation while maintaining throughput efficiency. b) Guaranteeing a stable and swift PCI-E bandwidth through the utilization of a hardware-topology-aware strategy. c) Decreasing host-side memory capacity overhead by leveraging continuous buffers. Furthermore, it is important to highlight that all these methods contribute to reducing $k$, which signifies the number of stages that can be offloaded. For a more detailed explanation of these techniques, please refer to Appendix C.

### 4.2. Offload Scheduling

When integrating offloading and reloading into a pipeline, careful scheduling alongside computation passes is essential. (Yuan et al., 2024) employs a fixed scheduling strategy, initiating offloading immediately after the forward pass and reloading at the start of the last backward pass. Offloading and reloading are placed into separate streams to enable overlap. However, in practice we find that separate streams

can lead to significant latency fluctuations (see Appendix D), which can result in notable overhead due to computation passes waiting for offloading or reloading to complete. In contrast, we use a single stream for both offloading and reloading. By sharing the stream, we stabilize the latency of offload and reload passes, simplifying the scheduling and enhancing system robustness and performance.

When scheduling the offload and reload passes based on uniform repeating strategy, as shown in Figure 7, we maintain pipeline bubbles after repeating the building block. For a given model and its training configurations, we first calculate $k$ as defined in Formula 1. Then, we allocate separate slots in the stream using a one-offload-one-reload pattern. Given the pipeline stages to offload, we process them one by one from left to right, finding the earliest available offload slot after the forward pass and placing the offload there. For reloading, we move from right to left, identifying the latest available reload slot before the backward pass and placing the reload there. Although some slots may remain unoccupied and the schedule is not squeezed, it can be optimized automatically when running on devices. For A100 GPUs, where PCI-E is often shared by two adjacent devices, we stagger the offload streams and insert synchronization events across the two devices to avoid the same operation occurs simultaneously (see Appendix D for more details).

### 4.3. $k$ on Other Hardware Platforms

In Figure 1 we showed that $k$ is relatively small on A100 GPUs. While the value of $k$ is reliant on the hardware architecture, we anticipate that H100 will exhibit a similar $k$ to A100, given that H100 boasts 2x PCI-E bandwidth following an upgrade from PCI-E v4 to v5, along with a 3x increase in compute bandwidth. Furthermore, the model flops utilization (MFU) typically reported by the community is lower for H100s (43% in Llama 3 (Dubey et al., 2024)) in contrast to A100s (approximately 60% in our evaluations), thereby narrowing the bounds even further.

### 4.4. Caveats of Offload

Though a carefully implemented offload strategy brings only negligible overhead to the compute throughput, there're some moderate issues. Firstly, the host memory capacity is another notable bound for the offload. However, host memory is usually several times larger than the total GPU memory installed on the host and is usually extensible with much lower cost compared to GPU. Secondly, achieving a "free lunch" offload scenario becomes unattainable if the time interval between matching forward and backward passes is shorter than the round-trip time of offload. A low time interval implies a short lifespan, leading to a negligible contribution to peak activation memory. In practice, skipping offloading for these passes has been observed not to im-

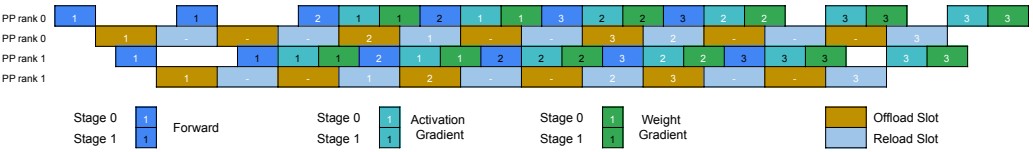

*Figure 7.* The offload scheduling based on uniform repeating with one pipeline stage offloaded per device and $k = 1$. We use a single stream with a one-offload-one-reload pattern, where "-" means empty slot.

pact peak activation memory. Lastly, the PCI-E traffic for data movement between host and device may interfere with cross-node P2P communication protocols such as Infiniband or RoCE, potentially slowing down communication if not scheduled prudently. Notably, P2P communication does not significantly impact the utilizable PCI-E bandwidh due to two reasons: a) The volume of P2P communication is substantially lower than offload. For each transformer layer, the total activation memory is ten times greater compared to the layer output. b) Most P2P communication occurs within a node on NVLINK, bypassing PCI-E, which mitigates the impact on offload constraints.

## 5. Experiments

We evaluate our methods on GPT-3-like models based on Megatron-LM (Narayanan et al., 2021). In most cases, one transformer layer is removed from both the first and last pipeline stages to address imbalances caused by vocabulary layers, similar to Llama 3 (Dubey et al., 2024) and Deepseek v3 (Liu et al., 2024). The models used are listed in Table 2. Our primary metrics are throughput, measured as model flops utilization (MFU), and activation memory, defined as the difference between peak and iteration-start memory. The reported activation memory refers to the maximum peak activation memory observed across all devices.

*Table 2.* A list of models used in experiments. For all models we turn on GQA (Ainslie et al., 2023) with number of query group set to 8.

| Model | Layers | Attention Heads | Hidden Size | Batch Size | GPUs |
|-------|--------|-----------------|-------------|------------|------|
| 5.8B  | 32     | 32              | 4096        | 32         | 2-32 |
| 10.5B | 38     | 40              | 5120        | 64         | 8    |
| 18.1B | 46     | 48              | 6144        | 128        | 16   |
| 42.9B | 62     | 64              | 8192        | 256        | 32   |
| 66.6B | 62     | 80              | 10240       | 256        | 32   |
| 83.8B | 78     | 80              | 10240       | 256        | 32   |

Our experiments run on up to 32 NVIDIA A100 80G GPUs on 4 nodes interconnected by RoCE RDMA network. As mentioned in Section 4.3, though the $k$ is hardware-dependent, we focus on A100 because it is similar for other

modern hardware, such as H100.

The pipeline schedules we evaluate include: a) *1F1B* (Harlap et al., 2018) and vanilla Interleaved 1F1B (referred to as *1F1B-I*) (Narayanan et al., 2021) implemented in Megatron-LM; b) Our generalized interleaved schedule, *GIS* and *GIS-H*, detailed in Section 3. c) Our better-than-linear offload techniques outlined in previous Sections. We concentrate on two key configurations, *PO-H* and *PO-F*, where either half($\lceil \frac{v}{2} \rceil$) or full($v$) stages are selectively offloaded. Notice that we skip the *PO-F* settings if the corresponding $k > 1$. For all schedules except *1F1B*, we set the number of stages on each device to the maximum possible value so that each stage has at most 1 transformer layer, unless explicitly specified.

### 5.1. Activation Memory Reduction with Similar Throughput

In Figure 8, we present a comparative analysis of activation memory and throughput across various methods. Compared to *1F1B-I*, our *GIS-H* method effectively halves the activation memory, while *PO-H* achieves a more substantial reduction, decreasing it to $\frac{1}{6}$ at most. *PO-F* further minimizes activation memory, providing a solution for scenarios where other methods might encounter out-of-memory issues.

In terms of throughput, all methods generally perform well, with pipeline bubbles only occurring during the warmup and cooldown phases. Our *GIS* method outperforms *1F1B-I* by offering both higher throughput and reduced activation memory. Although *PO-H* and *PO-F* show slightly reduced throughput compared to *GIS* and *GIS-H*, due to additional pipeline bubbles, they still surpass *1F1B* in throughput (except for cases which runs *PO-F* with $k$ exactly equals to 1). For detailed numerical results and explanations of any missing data points, please refer to Figure 15 in appendix.

### 5.2. Better-Than-Linear Selective Offload

In Figure 9, we illustrate the impact of memory savings when implementing stage-level offload across various schedules. The results indicate that our pipeline offload (PO) schedules with uniform repeating demonstrate memory savings that surpass linear scaling, unlike *GIS-H*, aligning with

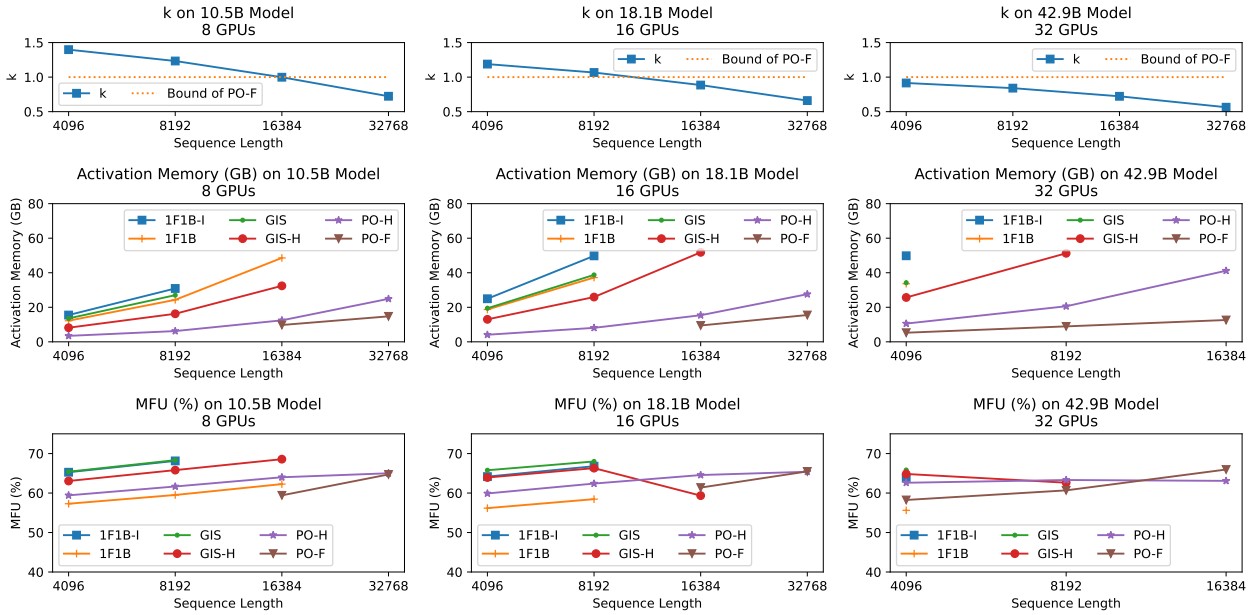

Figure 8. Memory and Throughput Comparison of Different Methods. For detailed numbers, please refer to Figure 15 in appendix.

the analysis in Section 2. It's important to note that across all schedules, there exists a consistent overhead caused by temporary activation memory when offloading all stages.

### 5.3. Activation Memory Scaling Study

We delve into the strong scaling of activation memory by analyzing per-device activation memory using a fixed 5.8B model across various total numbers of stages $v \times d$. The results depicted in Figure 10 reveal that the per-device activation memory (left figure) of *PO-H* and *PO-F* exhibits superior scaling compared to other methodologies, primarily due to the reduction in the number of in-flight activations (as indicated in the right figure). Notably, the number of in-flight activations remains constant for *PO-F*, matching the analysis in Table 1. This observation implies that in the most common scenarios where each stage has only 1 transformer layer, each device essentially maintains an activation memory equivalent to a small constant number (approximately 4 in our experiments) of transformer layers, irrespective of the total number of layers and pipeline devices.

### 5.4. Comparing With Tensor Parallelism

The high activation memory volume is one of the biggest concern for scaling PP to more devices. Commonly, standard settings such as Llama3 (Dubey et al., 2024) often employ a maximum TP degree, typically set at 8, to reduce the activation memory per device. By leveraging our methods to save activation memory on PP, we now compare the performance of using pure PP, which was previously unattainable without our techniques, with interleaved 1F1B combined with 8 TP (together with sequence parallelism in (Korthikanti et al., 2023)). The results depicted in Figure 11 showcase a notable 12%-19% acceleration in training, attributed to the elimination of TP, which typically incurs significant communication overhead. We also highlight that *PO-F* method not only exhibits higher throughput but also consumes less activation memory compared to *1F1B-I* combined with the maximum TP degree. This finding suggests that in scenarios where *PO-F* is applicable ($k \leq 1$), pure PP should be preferred.

### 5.5. Comparing With Other Pipeline Parallel Methods

We compare our approach with existing pipeline parallelism (PP) methods, particularly those aimed at reducing activation memory, including V-Min and V-Half from Qi et al. (2024), as well as activation re-materialization combined with 1F1B-I (shown as 1F1B-I-R). As shown in Figure 12, our methods achieve a more favorable Pareto frontier in the trade-off between memory usage and throughput. Through detailed analysis, we attribute the suboptimal performance of the V-Shaped schedule proposed in Qi et al. (2024) to significant disparities among $T_F$, $T_B$, and $T_W$ at large sequence lengths. These disparities violate the core assumption in Qi et al. (2024) that these timings are approximately equal, leading to degraded efficiency.

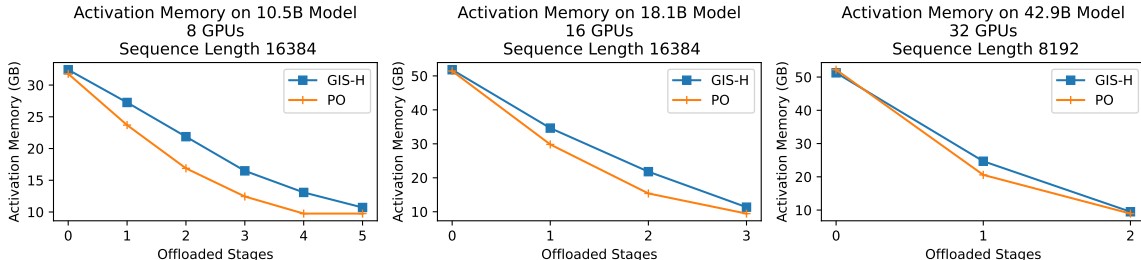

Figure 9. Better-than-linear selective offload. We gradually increase the number stages to offload on two schedules: *GIS-H* defined in Section 3 and Pipeline Offload (PO), the uniformly repeated schedule introduced in Section 2

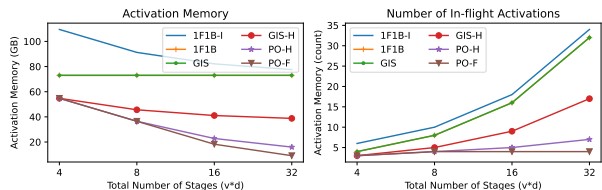

Figure 10. Per-device activation memory when training a 5.8B model using different total number of stages. The left figure shows the activation memory in GB while the right figure shows the number of in-flight microbatches (different stages of the same microbatch are counted multiple times). If there's multiple settings for the same $v * d$, the setting with minimum activation memory is reported. The amount of activation memory is estimated by running the scheduler and count the in-flight activation memory on GPU. Detailed data on this experiment is shown in Figure 18 in appendix.

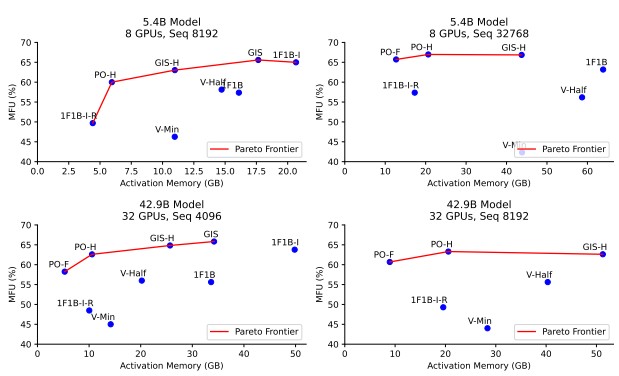

Figure 12. Pareto frontier compared with other methods.

### 5.6. Convergence Experiment

We compare the loss curves of our *PO-H* and *PO-F* implementations with 1F1B-I to validate correctness on a 5.4B model using 8 GPUs. As shown in Figure 13, the results

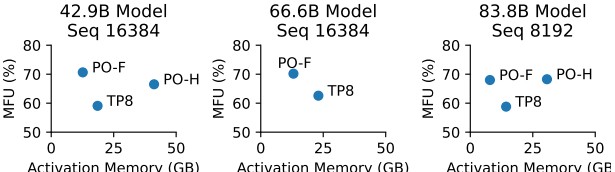

Figure 11. Comparison of pure PP using our methods with hybrid parallelism using PP+TP. *PO-F* and *PO-H* runs as 32-way pure PP while *1F1B-I* runs with TP8xPP4.

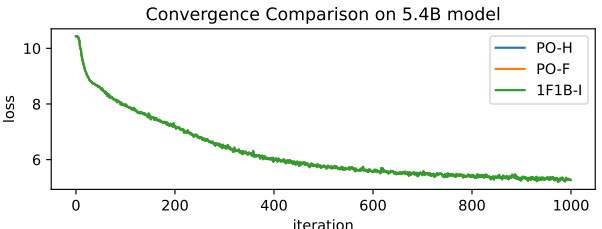

Figure 13. Loss curves proves algorithmic correctness of our methods.

exhibit nearly identical loss trajectories, providing strong evidence for convergence correctness. Notably, our methods are designed as exact algorithms with no compromises in accuracy.

## 6. Related Work

**Pipeline Parallelism**   Various pipeline schedules have been developed to reduce activation memory usage in PP. An early notable work is 1F1B (Fan et al., 2021; Harlap et al., 2018), which uses a one-forward-one-backward pattern to mitigate the high memory usage of GPipe (Huang et al.,

2019). BPipe (Kim et al., 2023) was later introduced to address the memory imbalance issue of 1F1B by transferring activations across devices. Leveraging the auto-regressive property of causal transformers, token-level pipeline schedules have been proposed by (Li et al., 2021; Sun et al., 2024), showing promising memory reduction results, especially for long-context training. Vocabulary parallelism was recently proposed by (Yeung et al., 2024) to address the memory imbalance caused by vocabulary layers, alleviating the memory bottleneck of PP. (Qi et al., 2024) introduced a general framework showing that peak memory can be directly controlled by lifespan of the building block. Based on this insight, they proposed a memory-balanced V-Shape schedule, reducing peak activation to at most 1/3 compared to 1F1B.

**Activation Rematerialization and Offload**  Activation rematerialization was first proposed by (Chen et al., 2016) which trades computation for memory. To alleviate its overhead, selective strategies have been developed (Korthikanti et al., 2023; Yuan et al., 2024), focusing on recomputing operations with high memory-to-computation ratio.

Offload techniques have also been explored to address memory constraints in training LLMs, but prior work (Ren et al., 2021; Rajbhandari et al., 2021) often focuses on model states, leading to poor overlap between data transfer and computation and resulting in high overhead. A recent work (Yuan et al., 2024) on offloading activation memory in PP is the most related work to ours. However they draw an opposite conclusion than ours, that activation offload causes significant overhead and should be avoided if possible. They emphasize memory reduction, allowing offload to delay computation, which often results in a brittle schedule introducing significant pipeline overhead. In contrast, we focus on improving the memory reduction efficiency and minimizing overhead by fully overlapping offload with computation. We deliver a different insight that offload can be a free lunch in PP, and full activation is often feasible to make PP scalable.

## 7. Conclusion

In this work, we present *PipeOffload*, a novel pipeline schedule that incorporates multiple innovative techniques to significantly decrease the activation memory requirements of PP. Through evaluation, we demonstrate that *PO-H* can reduce activation memory to less than a quarter of that required by interleaved 1F1B schedules while maintaining similar throughput across a wide range of real-world models. *PO-F* proves particularly impactful for larger models or extended sequence training tasks, where activation memory can be further reduced to that of a small constant number of transformer layers. These methods greatly improved the

scalability of PP and PP becomes feasible in cases they were not possible previously. We demonstrate that the enhanced PP can serve as a compelling alternative to other distributed training methods, such as tensor parallelism.

## Impact Statement

This paper presents work whose goal is to advance the field of Large Language Model Training Systems. There are many potential societal consequences of our work, none which we feel must be specifically highlighted here.

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

## A. 1F1B schedule with Fully Activation Offload

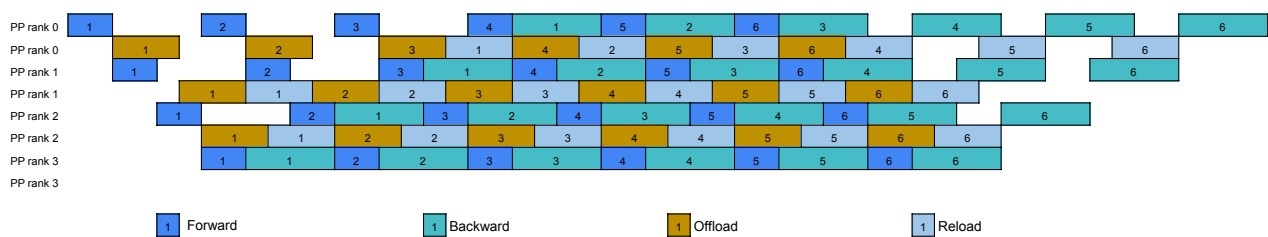

*Figure 14.* Schedule of applying fully activation offload to 1F1B. Note that in this schedule we also use the topology-aware offload which synchronizes offloading and reloading cross two adjacent devices.

## B. More Details on Memory and Throughput Comparison between Different Methods

In Figure 15, we show more details and numbers of the experiment introduced in 5.1

## C. Implementation

In this section, we show more implementation details on how we reduce the offload bound and save memory on both host and devices.

**Continuous Host Buffer Bins**  We notice that pytorch round up the size of host memory to nearest power of 2, resulting in at most 1x waste of host memory. To mitigate this for each offloading pass, we use bigger continuous buffer bins with sizes which are power of 2. We run a heuristic-based search to find a solution that **a)** All activation memory of a single offloading pass can fit these bins **b)** At most 3 bins are used **c)** The total size of the bins is minimum. In practice we find this method reduces the waste of host memory to a negligible amount. During reloading we move the continuous buffers directly to device and construct individual activation memory tensors on them.

**Deterministic Device Memory Management**  We allocate all offload-related buffers on the compute stream and use CUDA events to synchronize the usage of buffers between the offload stream and compute stream. This method circumvents the issue of non-deterministic buffer deallocation times that could prompt frequent cudaMallocs if one were to follow a more simplistic approach using Tensor.record_stream.

**Selective Recomputation with Negligible Overhead**  Using methods similar to (Chen et al., 2016), we simply recompute LayerNorm and GeLU layers in the backward pass to reduce the activation memory that is subject to offload. We also implement a customized dropout that preserves the random seed during the forward pass and reconstructs the dropout mask in the backward pass. Recomputing these layers reduces the activation memory for single layer from $34bsh$ (as (Korthikanti et al., 2023)) to $20bsh$. Our ablation study results demonstrate a 40% reduction in activation memory with negligible throughput overhead (around 1-2% performance impact). We conduct an ablation experiment as in Figure 16

**Topology-aware Offload Scheduling**  To achieve optimal performance, we implement a synchronized interleaved memory transfer schedule where two devices within the same NUMA node collaboratively execute alternating H2D and D2H transfers using cross-device event synchronization. The scheduling of memory transfer will be studied in Appendix D. Our approach carefully co-designs computation and memory transfer schedules to ensure transfer stability and efficiency while maintaining interdependencies.

**Enhanced Node Assignment**  In many parallel processing (PP) schedules, there exists an imbalance in activation memory across stages, leading to a linear decrease in activation memory from the initial to the final rank. This imbalance places varying loads on the host memory capacity of different nodes. To optimize host memory utilization, we consolidate the PP stages by grouping lower-ranked stages with higher-ranked stages on the same node. For instance, in a PP setup with 16 devices, ranks 0-3 and 12-16 are assigned to node 0, while ranks 4-11 are allocated to node 1. It is important to note that this approach results in a doubling of cross-node communications.

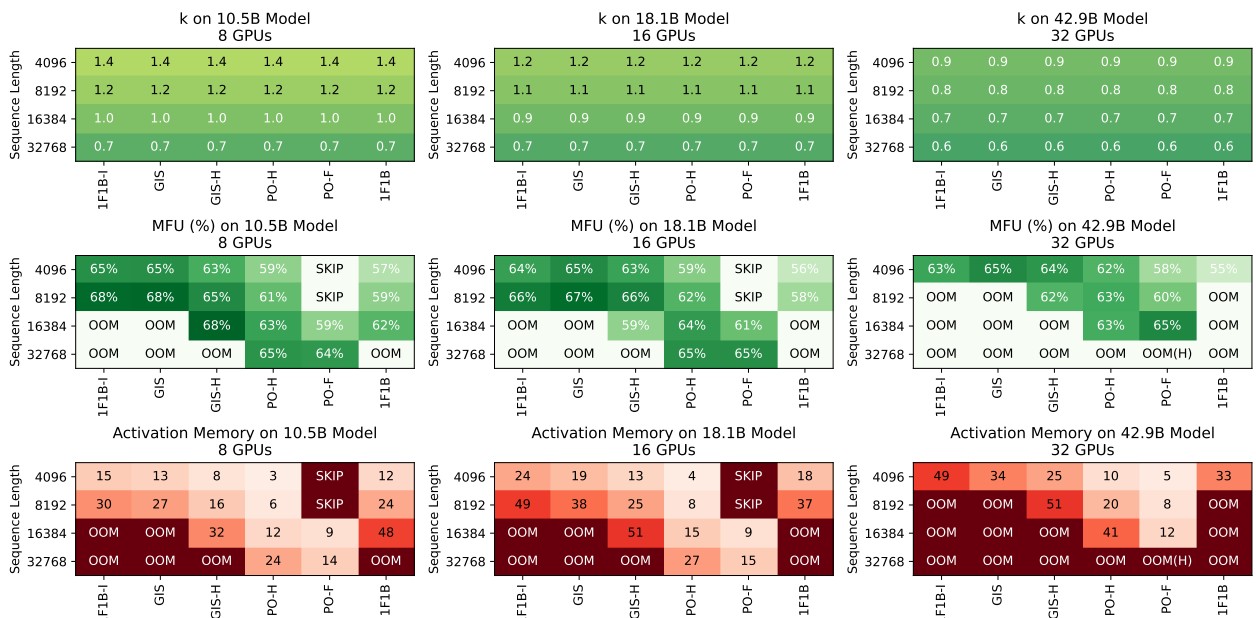

*Figure 15.* Detailed Data of Throughput and Memory Comparison of Different Methods. SKIP indicates the *PO-F* method is skipped because $k > 1$. While OOM indicates a GPU out-of-memory error, OOM(H) represents an OOM on host, which only happens on the largest model with largest sequence length.

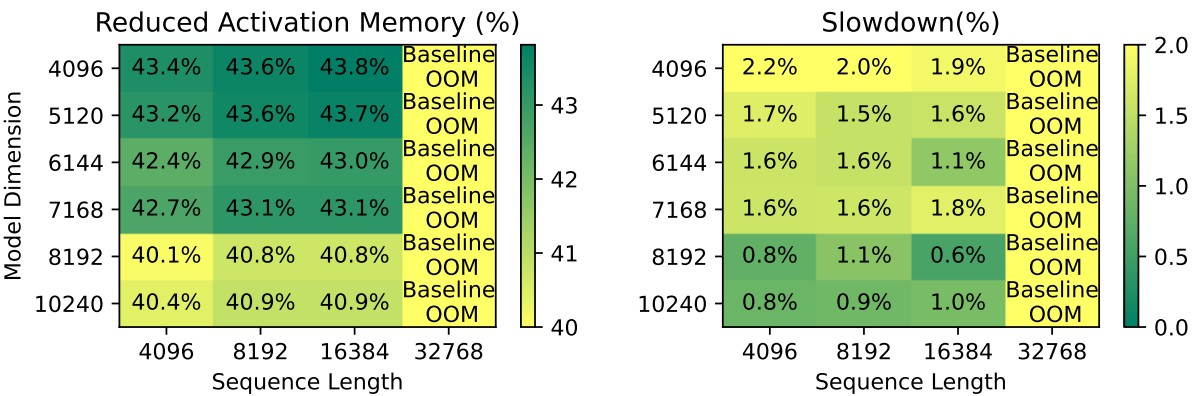

*Figure 16.* Ablation study on applying recomputation for Layernorm, GeLU and dropout layers. Approximately 40% activation is reduced with around 1-2% slowdown.

## D. Topology-aware Offload Scheduling

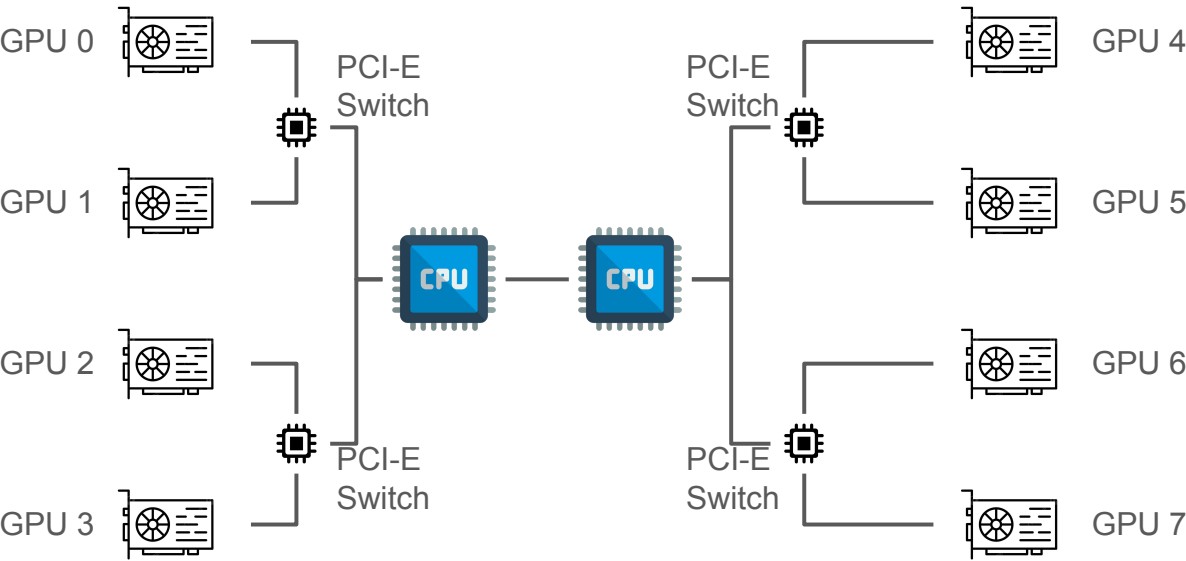

*Figure 17.* GPU server topology of our A100 GPU server. Adjcent GPUs share the same PCI-E switch hence potentially interfere with each other.

Designing the offload schedule based on the GPU server's topology is crucial. For instance, in our testing of the A100 server, illustrated in Figure 17, we observed that Host-to-Device (H2D) or Device-to-Host (D2H) transfers between GPU pairs on the same PCI-E switch could potentially interfere with each other. Therefore, coordinating the offload operations among GPU pairs on the same PCI-E switches becomes a critical consideration.

In our investigation, we assessed the throughput and stability of these transfers using various co-scheduling methods. We organized H2D and D2H operations into sequential groups, with operations within each group executing consecutively. To mimic real-world scenarios where two devices sharing the same PCI-E switch might trigger transfers at different times during training iterations, we introduced random inter-group delays. Each experiment was repeated multiple times, and we generated the distribution of per-device bidirectional throughput in Table 3.

It's worth noting that while our experimental results favor the synchronized interleaved method, server topologies can differ. Therefore, in practice, it's essential to select co-scheduling methods that are most compatible with a specific hardware configuration.

## E. Detailed Data on Activation Memory Scaling Study

In Section 5.3 and Figure 10, we've shown how activation memory scales with $v \times d$. In this section we show more detailed data on how the activation memory scale with $v$ and $d$ along, shown in a plot in Figure 18

*Table 3.* Comparing co-scheduling methods for offload operations on a pair of GPUs connected to the same PCI-E switch. Results shows that the syncronized interleaved schedule is both stable and fast.

| Schedule | Sketch | Bandwidth Histogram |
|---|---|---|
| **Parallel** Both devices under a NUMA node execute alternating H2D and D2H transfers parallelly. | | |
| **Interleaved** Similar to parallel H2D-D2H, but devices under each NUMA node begin with complementary operations. | | |
| **Dual Stream** Both devices simultaneously execute H2D and D2H operations using separate CUDA streams. | | |
| **Synchronized Parallel** Parallel H2D-D2H with cross-device event synchronization before each memory transfer. | | |
| **Synchronized Interleaved** Interleaved H2D-D2H with cross-device event synchronization before each memory transfer. | | |
| **Synchronized Dual Stream** Dual-Stream H2D-D2H with cross-device event synchronization before each memory transfer. | | |

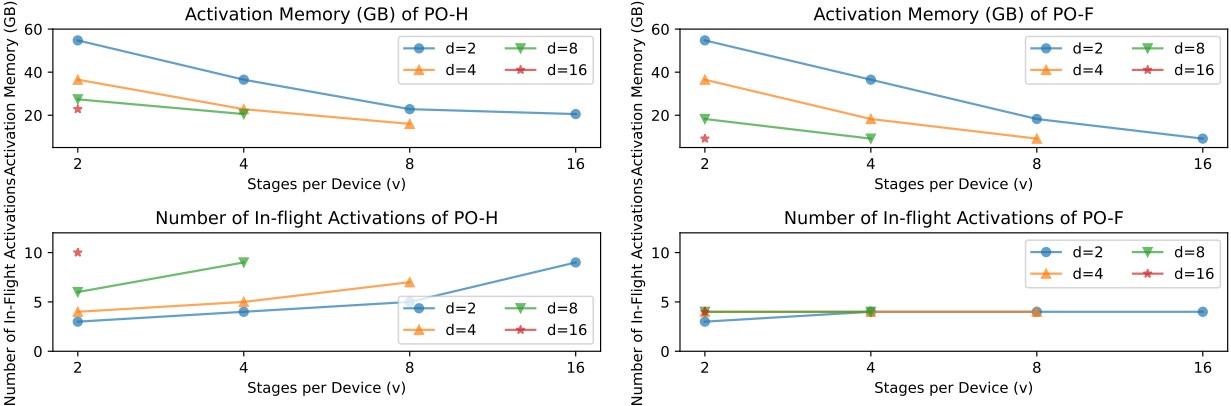

*Figure 18.* Scaling of activation memory under different $v$ and $d$.

