# OpenReview forum: "PipeOffload: Improving Scalability of Pipeline Parallelism with Memory Optimization"
_ICML.cc/2025/Conference — ICML 2025 poster_

### Official Review · Reviewer_7WYn · 2025-02-26

**Overall Recommendation:** 3

**Summary:**

In this paper, the authors present the memory offload strategies in pipeline parallel training. The authors suggest that the variables with long lifespan should be offload, and then introduce a selective offload strategy that decreases peak activation memory in a better-than-linear manner. Moreover, that offload strategy can jointly considered with other techniques. Experiments illustrate the offload strategy can introduce 19% acceleration with lower memory consumption.

**Claims And Evidence:**

To the reviewer's best knowledge, the claims and evidence are all supported by clear evidence. There is not any fatal mistake or overclaim in this paper.

**Essential References Not Discussed:**

NA

**Experimental Designs Or Analyses:**

The experiments can validate the benefits of the proposed offload strategies as the proposed method can achieve a better performance with the model size varying from 5.8B to 83.8B.

**Methods And Evaluation Criteria:**

The proposed method is effective in the pipeline parallel training according to the development of the method and the experimental results. It makes sense for the pipeline parallel training.

**Other Comments Or Suggestions:**

NA

**Other Strengths And Weaknesses:**

**Strength**
1. The writing is excellent. It is easy for readers who has little basic knowledge in pipeline parallelism to understand the manuscript.
2. The idea is simple but effective. It is easy to follow.
3. The authors take a thorough experiment under model with model size varying from 5.8B and 83.8B. And the proposed method can achieve a better performance.

**Weakness**
1. The offload strategy is only based on 1F1B strategy. It is unknown that how the proposed strategy in other pipeline parallel strategy like DualPipe [1].
2. The authors fail to provide the code of the proposed method, which may decrease the reliability of the experimental results.

[1] DeepSeek-V3 Technical Report

**Questions For Authors:**

1. Whether the proposed strategy is suitable for other pipeline parallel strategy like DualPipe? The authors can present some analysis theoretically and empirically.
2. Whether the authors can provide the core codebase of the experiments? It may make the experiments reproducible.

**Relation To Broader Scientific Literature:**

NA

**Theoretical Claims:**

The reviewer has taken a quick check on the theoretical claims. There is not any fatal issue.

---

> ### Author Rebuttal · Authors · 2025-03-31
>
> We thank the reviewer for the insightful feedbacks. See response to individual points below.
>
> > Whether the proposed strategy is suitable for other pipeline parallel strategy like DualPipe? The authors can present some analysis theoretically and empirically.
>
> TL;DR: Yes, but different devices will have different memory deduction.
>
> The selective offload strategy is applicable to bi-directional schedules like Chimera and DualPipe, as they involve multiple stages with different directions on the same devices. In these discussions, we refer to the stage on device $i$ with a forward pass from device $i$ to device $i+1$ as stage 0, and the reverse stage as stage 1.
>
> In DualPipe, stages on the same device have varying lifespans. For instance, on device 0, stage 0 has a longer lifespan, while stage 1 has a negligible one. This creates an opportunity for better-than-linear memory reduction, especially since DualPipe uses a "Uniform Repeat" strategy, as mentioned in Section 2.
>
> However, the lifespan distribution varies across devices. For example, stage $\frac{d}{2}$ has nearly equal lifespans for both stages, and selective offloading leads to linear memory reduction. In a rough analysis, offloading one stage on devices 0 and $d-1$ results in approximately 100% activation reduction, while on device $\frac{d}{2}$, it results in about 50%. The memory reduction for other devices can be linearly interpolated.
>
> Finally, we emphasize that DualPipe's main advantage is in expert-parallel (EP) pipeline scheduling, where it fuses forward and backward passes to overlap EP communication with computation.
>
> > Whether the authors can provide the core codebase of the experiments? It may make the experiments reproducible.
>
> We are going to open-source the code base for our implementation and experiments for a broader impact in the community. We did not upload it as supplementary material because we were not confident in anonymizing the code sufficiently to comply with double-blind review rules.

---

### Official Review · Reviewer_UJJy · 2025-03-12

**Overall Recommendation:** 4

**Summary:**

This paper proposes PipeOffload, a novel selective offload strategy to decrease peak activation memory in pipeline parallelism training. PipeOffload greatly improves the scalability of pipeline parallelism, and makes pipeline parallelism a stronger alternative than tensor parallelism.

**Claims And Evidence:**

Claims are supported by clear and convincing evidence.

**Essential References Not Discussed:**

No.

**Experimental Designs Or Analyses:**

The experimental design is sound.

**Methods And Evaluation Criteria:**

Methods and evaluation make sense.

**Other Comments Or Suggestions:**

No.

**Other Strengths And Weaknesses:**

Strengths
1. The paper is well-written and clearly motivated.
2. Address an important problem.
3. The paper presents a novel approach to reduce peak memory consumption in PP and achieves good performance.

Weaknesses
1. The selective offload strategy introduces additional complexity in managing activation lifespans, which could complicate implementation and maintenance.
2. The effectiveness of the proposed strategies may vary across different hardware configurations, potentially limiting their general applicability.

**Questions For Authors:**

1. How seamlessly can PipeOffload be integrated into existing machine learning frameworks that utilize 3D parallelism (combining data, tensor, and pipeline parallelism), and what challenges might practitioners face during such integration?
2. Given that PipeOffload introduces a higher bubble overhead compared to the 1F1B-I (One-Forward-One-Backward Interleaved) scheduling strategy, is it feasible to apply PipeOffload in conjunction with 1F1B-I to mitigate this issue?
3. In Figure 7 of your paper, how did you determine the durations for offloading/reloading and forward/backward computations? Could you elaborate on the methodology used to decide these lengths?
4. Can you elaborate on the memory imbalance across different stages? Does PipeOffload need to offload/reload the same amount of activations for different stages?
5. How does PipeOffload's performance compare to that of bidirectional pipeline parallelism? Is it possible to combine PipeOffload with bidirectional pipeline parallelism strategies?
6. How to decide the offload schedule for different topologies?

**Relation To Broader Scientific Literature:**

This paper helps improve the LLM training efficiency with pipeline parallelism.

**Theoretical Claims:**

No.

---

> ### Author Rebuttal · Authors · 2025-03-31
>
> We thank the reviewer for the insightful feedbacks. See response to individual points below
>
> > The selective offload strategy introduces additional complexity in managing activation lifespans, which could complicate implementation and maintenance.
>
> Every new feature requires some implementation, however, we try our best to make implementation simple. The offloading majorly leverages the existing PyTorch ```Tensor.to()``` function to do D2H and H2D transfers. The scheduling code handles offloading as PP native passes and uses simple heuristics to determine the ordering between them.
>
> > The effectiveness of the proposed strategies may vary across different hardware configurations, potentially limiting their general applicability.
>
> We agree that the offload time ratio $k$ is dependent on hardware, particularly the host-to-device bandwidth. Our implementation allows for adjustable scheduling to accommodate different hardware configurations.
>
> Additionally, mainstream GPUs are rapidly improving in PCIe bandwidth (e.g., A100: 64 GB/s, H100: 128 GB/s, B100: 256 GB/s, Grace Hopper: 900 GB/s). Given this trend, we expect our method to remain generalizable to future devices.
>
> > How seamlessly can PipeOffload be integrated into existing machine learning frameworks that utilize 3D parallelism
>
> Our implementation is based on Megatron-LM, one of the most popular LLM training frameworks. PipeOffload is orthogonal to other parallelism schemes and requires minimal additional effort when integrated into a hybrid parallelism strategy.
>
> > is it feasible to apply PipeOffload in conjunction with 1F1B-I to mitigate this issue?
>
> Yes, the same method can be applied to 1F1B-I. However, as discussed in Section 2 of our paper, selective offloading in 1F1B-I only achieves linear memory savings. This setup is still preferable when higher throughput is the priority.
>
> > in Figure 7 of your paper, how did you determine the durations for offloading/reloading and forward/backward computations? Could you elaborate on the methodology used to decide these lengths?
>
> In our current implementation, for simplicity, we use the ratio $k$ between offload time and compute time, as calculated by Formula 1, while assuming F/B/W passes run for identical time. Experiments show that this approach works well. However, we anticipate that using more accurately profiled timings for compute and offload passes will lead to even better performance.
>
> > Can you elaborate on the memory imbalance across different stages? Does PipeOffload need to offload/reload the same amount of activations for different stages?
>
> Could you clarify whether this imbalance refers to the different activation memory for different stages on the same device, or different stages on different devices? If it's about different stages on the same device, the imbalance arises from the varying lifespans of activations across stages. For example, in the forward pass, stage 0 computes before stage 1, while the backward pass is computed after stage 1. This results in stage 0’s activations living longer than those of stage 1, contributing to higher peak memory for stage 0.
> If multiple stages are offloaded, the number of activations to be offloaded will be the same across stages, but the peak memory reduction will differ, with stage 0 providing a greater reduction than stage 1.
>
> > How does PipeOffload's performance compare to that of bidirectional pipeline parallelism? Is it possible to combine PipeOffload with bidirectional pipeline parallelism strategies?
>
> The same insight of selective offloading can also be applied to bidirectional PP, as in bidirectional PP there're also multiple stages on the same device.
>
> > How to decide the offload schedule for different topologies?
>
> The topology influences how offload/reload passes are co-scheduled across different devices. For instance, if two GPUs are connected to the same PCIe switch, we schedule them in a "Synchronized Interleaved" pattern, as described in Table 3. If the GPUs have independent PCIe connections, we schedule offload/reload passes independently on each device.

---

### Official Review · Reviewer_bsbd · 2025-03-14

**Overall Recommendation:** 3

**Summary:**

This paper proposes a method for optimizing large language model (LLM) training by leveraging pipeline parallelism (PP) focusing on selectively offloading activations. It identifies the opportunity to overlap computing and data transfer during the forward and backward passes, and proposes selective offloading based on layer lifespan. The authors compare several schedules (1F1B, 1F1B-I, GIS, GIS-H, PO-H, and PO-F) analyzing memory/throughput trade-offs. The experiments demonstrate improvements in activation memory and Model FLOPS Utilization (MFU) compared to other PP methods, or 12–19% training speed-up compared to tensor parallelism (TP) when using the Megatron-LM model and scaling across 2–32 A100 80G GPUs.

**Claims And Evidence:**

I do not notice any issue regarding unsupported claims. For example, the claim about selective offload is well supported by both theoretical prediction and experimental measurements.

**Essential References Not Discussed:**

N/A

**Experimental Designs Or Analyses:**

I reviewed the experimental designs and analyses and, while I did not identify any immediately apparent issues, I am not fully confident whether they are sound. Please find my questions below.

**Methods And Evaluation Criteria:**

It would be helpful to further improve the evaluation by considering these issues:
1. The acceleration factor is only measured against tensor parallelism methods instead of other pipeline parallelism methods.
2. Given pipeline parallelism being a research hotspot, expanding the comparison baselines to include methods like BPipe, GPipe, and PipeFill could provide valuable benchmark comparisons and strengthen the paper's claims if any of them is open-source.

**Other Comments Or Suggestions:**

1. Introducing abbreviations (e.g., 1F1B, GIS, ZB-H2) before their first usage would enhance readability.
2. It would be beneficial to explain why Megatron-LM is selected as the evaluation target if that is not common knowledge.
3. Minor typo: Line 327, “similar to.”

**Other Strengths And Weaknesses:**

+ The paper includes vibrant figures to help understanding.
- The design section could benefit from clarifying the originality of each optimization technique discussed.

**Questions For Authors:**

Thanks for submitting to ICML’25. This is solid work and I generally enjoyed my reading. My overall rating of this paper is somewhat neutral. It would be extremely helpful if the authors could properly address these questions:

Novelty:

1. It would be helpful to address the fundamental differences against the existing work “Pipeline Parallelism with Controllable Memory. Qi et al, 2024.”
2. Is GIS(-H) a realization of an existing method or an entirely new design? If existing, outlining the differences between PipeOffload and GIS(-H) would be helpful. If new, adding one more state-of-the-art baseline in the evaluation might be worth considering.

Evaluation:

3. Regarding the main evaluation, what is the acceleration of PipeOffload compared to state-of-the-art *pipeline* parallelism methods (rather than tensor parallelism ones)?

Technical details:

4. Regarding Figure 2, what is the cause of the asymmetry between Stages 0 and 1?
5. Regarding Figure 3, how is the curve modeled and calculated? What does the kink near x=13 mean?

**Relation To Broader Scientific Literature:**

The paper lacks clarity regarding its novel contributions relative to existing research. Specifically, if this paper includes a concise listing of key contributions, I would be able to determine its novelty more precisely.
Furthermore, the distinction between established and novel concepts, such as Selective Offload and Zero-Bubble Strategy, is not adequately delineated. The paper should clearly identify which aspects are pre-existing and which are original, and elaborate on their differences.

**Theoretical Claims:**

The equations presented in Section 3 appear reasonable to me.
It would be advantageous to also cross-validate certain equations, for example, memory and bubble amount, against experimental measurements.

---

> ### Author Rebuttal · Authors · 2025-03-31
>
> We thank the reviewer for the valuable comments. We respond to individual points from your review below.
>
> > The paper lacks clarity regarding its novel contributions ... the distinction between established and novel concepts, such as Selective Offload and Zero-Bubble Strategy, is not adequately delineated ...
> > The design section could benefit from clarifying the originality ...
>
> Our main contributions are:
> * For the first time, we identify the oppotunity that all activation memory of PP can be offloaded with negligible overhead, which is critical but overlooked for the scalability of PP.
> * We propose a novel selective offload strategy in PP, which can achieve better-than-linear memory reduction. To the best of our knowledge, this is the first selective strategy for offload.
> * We extend the popular interleaved 1F1B into a generalized form for the first time, which can smoothly trade memory with minimal throughput loss, providing a flexible solution that can be tailored to specific system requirements.
> * We provide implementation details in Section 4 and Appendix, which is critical to achieve high performance for offload in PP.
>
> As referenced in the paper, Zero-Bubble was proposed by [1]. We apply this strategy as a preliminary step in our generalized interleaving strategy (Section 3.2). We don't claim it as our contribution, as it's trivial.
>
> > It would be helpful to address the fundamental differences against the existing work [2]
>
> Although we are inspired by some concepts (building-block, lifespan) from [2], our work has fundamentally different contributions and insights compared to [2]. Specifically, we focus on selective offloading (Section 2), designing a novel scheduling methodology to facilitate offloading (Section 3), and implementing optimized offload techniques (Section 4). These contributions are centered on offloading, whereas [2] aims to reduce activation memory through schedule-based methods, and does not mention offloading at all.
>
> > Is GIS(-H) an existing method or an entirely new design? ...
>
> The generalized interleaving strategy (Section 3.2) is entirely new. Both GIS and GIS-H can be seen as special cases of tuning $g$ to $g=d$ or $g=\frac{d}{2}$. While GIS happens to resemble Figure 17\(c\) in [2], it is not identical. To the best of our knowledge, GIS-H ($g=\frac{d}{2}$) and all GIS schedules with $g \lt d$ are entirely novel.
>
> > acceleration factor is only measured against TP instead of other PP methods.
> > what is the acceleration of PipeOffload compared to SOTA PP methods (rather than TP ones)?
> > Given PP being a research hotspot, expanding the comparison baselines to include methods like BPipe, GPipe, and PipeFill ... if any of them is open-source.
>
> Our primary focus is memory reduction to expand the applicability of pipeline parallelism (PP). Like all memory-saving methods, ours does not improve PP throughput; rather, it trades throughput for memory savings. This makes comparisons with tensor parallelism (TP) relevant, as TP is often preferred due to PP’s high memory requirements.
>
> BPipe and [2] are important works sharing the same target of memory saving with us, while only [2] is open-source. We conducted experiments comparing our approach with [2] and 1F1B-I with rematerialization which are the SoTA memory saving PP schedule. The results, shown in the image below, demonstrate that our methods (PipeOffload and GIS schedules) push the Pareto frontier of memory and throughput.
>
> [Pareto Frontier Figure](https://i.imgur.com/GiQXeSB.png)
>
> In this figure, 1F1B-I-R represents Interleaved 1F1B with full rematerialization, while V-Min and V-Half are from [2].
>
> > Regarding Figure 2, what is the cause of the asymmetry between Stages 0 and 1?
>
> Are you asking why the trends of Stage 0 and Stage 1 are reversed? Intuitively, after merging both stages, the final schedule follows a one-forward-one-backward pattern, ensuring a constant total activation memory across the two stages. As a result, when the memory usage of one stage increases, the other decreases.
>
> > Regarding Figure 3, how is the curve modeled and calculated? What does the kink near x=13 mean?
>
> We measure the percentage of memory reduction at $x \in \{0, 1, \ldots, 16\}$ and connect these points to form a curve. The kink near $x=13$ is simply because the evaluation points are discrete ($x$ must be an integer).
>
> > ... It would be advantageous to also cross-validate certain equations, for example, memory and bubble amount ...
>
> As mentioned in the caption of Figure 8, we provide detailed numbers in Figure 13.
>
> >It would be beneficial to explain why Megatron-LM is selected as the evaluation target if that is not common knowledge.
>
> We believe it's a common knowledge that Megatron-LM is SOTA and a common choice.
>
> [1] Qi, P., Wan, X., Huang, G., and Lin, M. Zero bubble pipeline parallelism.
>
> [2] Qi, P., Wan, X., Amar, N., and Lin, M. Pipeline parallelism with controllable memory.

---

### Official Review · Reviewer_s5WZ · 2025-03-14

**Overall Recommendation:** 3

**Summary:**

In this work, the authors introduced PipeOffload, a pipeline parallelism strategy that offloads activations with negligible overhead. The evaluation results demonstrate that PipeOffload uses less memory than the 1F1B method, enabling LLM training with larger model sizes and longer sequence lengths.

**Claims And Evidence:**

The evidence provided is generally sufficient to support the claims.

**Essential References Not Discussed:**

The provided references are appropriate and sufficient. However, incorporating comparisons with recent pipeline parallelism approaches, such as Chimera and DualPipe, could further strengthen the work and enhance its impact.

**Experimental Designs Or Analyses:**

The evaluation is generally satisfactory. However, including learning curves could enhance the understanding of convergence and correctness.

**Methods And Evaluation Criteria:**

The chosen baselines and datasets are generally appropriate for the study.

**Other Comments Or Suggestions:**

The proposed method seems more competitive for training larger models or longer sequences with limited memory. Other methods like 1F1B-I may achieve better MFU if sufficient memory is available.

**Other Strengths And Weaknesses:**

1. It is unclear how bandwidth affects the performance of the proposed method. What is the minimum memory bandwidth required to hide the offload overhead? Is this method applicable on systems with only PCIe?

2. Additionally, it is unclear in what case the proposed pipeline parallelism method can outperform tensor parallelism.

3. In modern LLM training, hybrid parallelism is commonly employed, combining data parallelism, tensor parallelism, pipeline parallelism, and sequence parallelism. Since using pipeline parallelism alone is uncommon, further discussion may be needed on how this method can be integrated with other parallelism strategies to ensure efficient scalability and performance.

**Questions For Authors:**

N/A

**Relation To Broader Scientific Literature:**

This work introduces a novel pipeline parallelism approach that reduces memory usage without additional overheads. This is significant for modern LLM training and could impact parallelization strategy design.

**Theoretical Claims:**

The theoretical claims appear to be correct.

---

> ### Author Rebuttal · Authors · 2025-03-31
>
> We thank the reviewer for the insightful feedbacks. Please see individual points below.
> > Including learning curves could enhance the understanding of convergence and correctness.
>
> We checked that our methods produces identical loss curves compared to vanilla 1F1B-I. We'll include this in the PDF.
> [Convergence Graph](https://i.imgur.com/bVgxTQb.png)
>
>
> > Incorporating comparisons with Chimera and DualPipe
>
> Our primary contribution is memory savings with minimal overhead. Since DualPipe, Chimera, and 1F1B share identical peak memory consumption, comparing against 1F1B and 1F1B-I sufficiently supports our claim.
>
> Additionally, DualPipe's advantage lies in expert-parallel (EP) pipeline scheduling, where it fuses forward and backward passes to overlap EP communication with computation. As our work does not focus on EP integration, a comparison with DualPipe with EP is beyond our scope.
>
> Moreover, we have added experiments comparing our approach with state-of-the-art memory-saving methods, including the V-Shape PP schedules from [2] and 1F1B-I with rematerialization (recompute). The results, shown in the image below, demonstrate that our methods push the Pareto frontier of the throughput-memory trade-off.
>
> [Pareto Frontier Figure](https://i.imgur.com/GiQXeSB.png)
>
> In this figure, 1F1B-I-R represents Interleaved 1F1B with full rematerialization, while V-Min and V-Half are from [2].
>
>
> > It is unclear how bandwidth affects the performance of the proposed method. What is the minimum memory bandwidth required to hide the offload overhead? Is this method applicable on systems with only PCIe?
>
>
> Common hardware, including PCIe-only systems, already provides sufficient bandwidth for activation offloading. Our method remains effective under these conditions.
>
> We extensively analyzed this in the introduction section. Formula 1 and Figure 1 demonstrate that typical hardware can offload a large portion (at least half, possibly all) of activations. In Formula 1, $k$ describes the ratio between offload time and compute time, and we see an important term $B_o$ which is the offload bandwidth. We plotted practical $k$ values based on A100 GPUs with PCIe v4 on Figure 1 (left) and have shown that a $k$ is actually small. Section 4.3 further discusses $k$ across different hardware, confirming similar trends.
>
> Notably this only relies on the host-device PCIe bandwidth, so it also works on systems with only PCIe.
>
> To clarify this emphasis on bandwidth, we will update the PDF to highlight the role of Formula 1 and Figure 1.
>
>
> > it is unclear in what case the proposed pipeline parallelism method can outperform tensor parallelism.
> > Discussion on being integrated with other parallelism strategies
>
> Thanks for raising this important discussion on TP vs. PP. In short, PP should be preferred when memory allows, as it generally provides higher throughput.
>
> Our results in Figure 11 align with prior work [1][2], showing PP outperforms TP due to TP’s high communication overhead. However, when memory is limited, TP can still help reduce activation memory. This highlights our contribution—reducing PP memory usage so that less TP is needed.
>
> On implementation, PipeOffload is orthogonal to other parallelization strategies and can be integrated into hybrid parallelism.
>
>
> A summary of a detailed analysis of TP vs PP that we'll add to the PDF:
>
> - Activation Memory: When using $d$ devices, For TP, each GPU has approximately $\frac{1}{d}$ of the activation memory of the entire model. For our PipeOffload, with selective offloading each device consumes approximately $\frac{1}{4}$ for a selective offload and $O(\frac{1}{vd})$ for a fully offload. Considering in practical scenarios $d\le8$ for TP, PipeOffload has a similar, or even situationally (when $k\lt1$, see Figure 1 in our paper) lower per-device activation memory compared to TP.
> - Throughput: Prior studies [2][3] show that TP with 8 devices incurs a 20–30% throughput overhead. For PipeOffload, the bubble rate is $\lt\frac{d-1}{n+d-1}$, where n is the number of microbatches. In most practical cases, the bubble rate of PipeOffload is lower than the overhead of TP.
>
>
> > Other methods like 1F1B-I may achieve better MFU if sufficient memory is available.
>
> We agree that when memory is sufficient, throughput should be prioritized. However, in many practical cases, "sufficient memory" is only achieved by using TP, which incurs significant throughput overhead, or by applying activation rematerialization. Our work provides an effective alternative for acceleration in such scenarios.
>
> References
>
> [1] BPipe: Memory-Balanced Pipeline Parallelism for Training Large Language Models
>
> [2] Pipeline Parallelism with Controllable Memory
>
> [3] Megatron-LM: Training Multi-Billion Parameter Language Models using GPU Model Parallelism.

---

### Decision · Program_Chairs · 2025-05-01

**Decision:**

Accept (poster)

**Comment:**

This paper proposes a new pipeline parallelism approach for efficient training. Concretely, it proposes to selectively offload activations to the host such that the latency from data movement can be effectively hidden. On a 32 A100 node setup, this approach makes pipeline paralleism an attractive alternative to tensor parallelism across various model sizes.

This is a solid paper that presents a sensible method backed up by empirical experiments, and thus I am recommending that this paper be accepted. (It would have been nice to consider training setups with even a greater number of GPUs, though I acknowledge that this may not be feasible on an academic budget.)